# Clinical and Molecular Insights of Radiation-Induced Breast Sarcomas: Is There Hope on the Horizon for Effective Treatment of This Aggressive Disease?

**DOI:** 10.3390/ijms23084125

**Published:** 2022-04-08

**Authors:** Stefania Kokkali, Jose Duran Moreno, Jerzy Klijanienko, Stamatios Theocharis

**Affiliations:** 1First Department of Pathology, Medical School, National and Kapodistrian University of Athens, 75 Mikras Asias Street, 11527 Athens, Greece; stamtheo@med.uoa.gr; 2Oncology Unit, 2nd Department of Medicine, Medical School, National and Kapodistrian University of Athens, Hippocratio General Hospital of Athens, V. Sofias 114, 11527 Athens, Greece; 3Hellenic Group of Sarcoma and Rare Cancers, G. Theologou 5, 11471 Athens, Greece; duranmoreno.jose@gmail.com; 4Department of Pathology, Institut Curie, 26 Rue d’Ulm, CEDEX 05, 75248 Paris, France; jerzy.klijanienko@curie.fr

**Keywords:** breast sarcoma, angiosarcoma, radiation-induced sarcoma, radiotherapy

## Abstract

Radiation-induced breast sarcomas (RIBS) are rare entities representing <1% of all primary breast malignancies, limiting most reports to small retrospective case series. They constitute a heterogeneous group of neoplasms, with high-grade angiosarcoma being the most common subtype. Other sarcoma histotypes, such as undifferentiated pleomorphic sarcoma and leiomyosarcoma, can also be identified. Radiation-induced breast angiosarcoma (RIBA) has an incidence of approximately 0.1% after breast-conserving therapy and arises mainly from the dermis of the irradiated breast. MYC gene amplification is highly indicative of secondary breast angiosarcomas. Their clinical presentation often mimics benign port-radiation lesions, leading to a delay in diagnosis and a lost window of opportunity for cure. Surgery with negative margins is the mainstay of treatment of localized RIBS. In the case of angiosarcoma, technical difficulties, including multifocality, infiltrative margins, and difficulty in assessing tumor margins, render surgical treatment quite challenging. A limited number of studies showed that adjuvant radiation therapy reduces local recurrences; therefore, it is proposed by many groups for large, high-grade tumors. Chemotherapy has been evaluated retrospectively in a small subset of patients, with some evidence supporting its use in angiosarcoma patients. Approximately half of patients with RIBA will show local recurrence. In the advanced setting, different therapeutic options are discussed in the review, including chemotherapy, antiangiogenic therapy, and immunotherapy, whereas the need for further research on molecular therapeutic targets is pointed out.

## 1. Introduction on Breast Sarcomas

Breast cancer represents the most common cancer in women and the second most common cancer-related cause of death [1]. The vast majority of invasive breast cancers are adenocarcinomas, arising from the terminal duct lobular unit. Breast sarcomas (BS), on the other hand, are very rare histologically heterogeneous nonepithelial malignancies, arising from the mesenchymal breast tissue. They should be differentiated from malignant phyllodes tumor and metaplastic breast carcinoma [2]. The rarity of such tumors limits most studies to small retrospective case series and case reports, most of which have limited long-term follow-up.

The median age at diagnosis of BS varies between 45 and 55 years [3,4,5,6]. The median size of the primary tumor is approximately 5 cm, which is larger than epithelial breast carcinoma (BC), whereas nodal involvement is very uncommon. The most common subtypes of BS are angiosarcoma, stromal sarcoma, liposarcoma, fibrosarcoma, osteosarcoma, chondrosarcoma, leiomyosarcoma, undifferentiated pleiomorphic sarcoma, and Kaposi sarcoma [7,8,9], with the most common being angiosarcoma. Factors identified to have a prognostic significance, associated with either overall survival (OS) or disease-free survival (DFS), are tumor grade, angiosarcoma histology, surgical margins, and tumor size.

A clear distinction exists between the de novo developing primary breast sarcomas (PBS) and secondary breast sarcomas (SBS) developing after radiation therapy (RT), or in the setting of chronic lymphedema, similarly to secondary sarcomas of the arm (Stewart-Treves syndrome) after axillary lymph node dissection [6,10,11,12]. Both are rare, representing <1% of all primary breast malignancies and <5% of all sarcomas [13]. The annual incidental rate for breast sarcoma varies in the literature according to the histotypes included in the different series. A retrospective analysis of the Swedish cancer registry identified an overall incidence of 1.5–2 cases per million per year [14]. Radiation-induced BS (RIBS) comprises approximately one-third of BS, with angiosarcoma being the most common subtype. The cumulative incidence of RIBS was estimated at 0.3% at 15 years post-RT [15]. Karlsson et al. reported a standardized incidence ratio of 1.9 for BS in women treated for BC [16].

The modified criteria, proposed by Cahan et al. [17] for radiation-induced bone sarcomas, are also used to define RIBS and include: different histological features between the primary tumor and the present sarcoma; the development of sarcoma in a previously irradiated field; a latent period typically >5 years; and histological confirmation of the second malignancy as sarcoma. These criteria were further modified by Arlen et al. in 1971, including tissues adjacent to the irradiated field and a shorter latency period (3–4 years) [18]. The latency period is necessary to distinguish RIBS from PBS, but the minimum required interval has been controversial among authors [19,20].

Common breast imaging studies, including mammography and ultrasound, are not specific for BS in demonstrating a suspicious mass. Fine-needle aspiration cytology is not very helpful, as adequate sampling is required for accurate preoperative diagnosis. Therefore, core biopsy is the gold standard, coupled with histological examination by an expert sarcoma pathologist, to distinguish BS from metaplastic carcinoma and malignant phyllodes tumor [21]. Breast magnetic resonance imaging (MRI) is indicated for planning appropriate surgery. The staging system of the American Joint Committee on Cancer/International Union Against Cancer is the most commonly used, just as for soft tissue sarcomas (STS) arising in other sites.

In the current review, the clinical and histopathological characteristics of RIBS will be discussed, as well as their management and prognosis, emphasizing their distinct characteristics versus PBS.

## 2. Histological Findings in Radiation-Induced Breast Sarcomas—Focus on Angiosarcoma

BS is a heterogeneous group of tumors, with angiosarcoma comprising approximately 50% of the cases of RIBS, corresponding to the most common histotype [12,15,22]. Nevertheless, other histotypes have also been described, including undifferentiated pleomorphic sarcomas (formerly referred to as malignant fibrous histiocytomas), leiomyosarcomas, fibrosarcomas, osteosarcoma etc. [11,19,23,24,25]. Figure 1 presents cytologic and histologic findings of radiation-induced osteosarcoma of the breast. In this case, cytological smears showed clustered or isolated spindle and pleomorphic sarcomatous cells associated with a pinkish/orange osteoid. Corresponding surgical sections showed a spindle-shaped and epithelioid proliferation with a foci of osteogenesis. Most of the reported cases of RIBS were diagnosed before 2010, and a large number of them before 2000, when diagnostic molecular techniques were not available. Therefore, accurate histological classification is questioned.

Angiosarcomas are rare but aggressive tumors that account for less than 5% of all soft tissue sarcomas (STS). They arise from the endothelial cells of blood vessels (hemangiosarcoma) or lymphatics (lymphangiosarcoma), either sporadically (as primary neoplasms) or secondary to chronic lymphedema and previous irradiation. In this review, only breast hemangiosarcoma will be discussed. Primary breast angiosarcoma (PBA) occurs in breast parenchyma, whereas radiation-induced breast angiosarcoma (RIBA) involves mainly the dermis and may or may not infiltrate breast parenchyma. Contrary to Stewart–Treves angiosarcoma, RIBA usually lacks lymphedema changes [26].

Before RT became a common practice for breast cancer, treatment-related angiosarcomas were usually lymphangiosarcomas, arising in the lymphedematous breast and upper arm following mastectomy and axillary dissection, and associated with chronic lymphedema [27]. The introduction of breast conserving surgery (BCS) and sentinel lymph node sampling has led to a decrease in treatment-related lymphedema and consequent lymphangiosarcoma, while at the same time, to an increase in the incidence of RIBA.

RIBA typically exhibits a vasoformative, and less commonly, a solid growth pattern [26,28]. An analysis of 27 cases of RIBA demonstrated a vasoformative pattern of growth combined with sieve-like or solid pattern areas, mainly high-grade nuclear features (16 grade 3, 8 grade 2, 3 grade 1), high mitotic rate, and rarely, necrosis [26]. The predominance of high-grade angiosarcoma is reported in all studies of RIBA with available histological data [29,30,31,32,33]. Furthermore, RIBA is more often multi-focal, and the macroscopic aspect is shown in Figure 2. Cytologic findings and corresponding histological sections are shown in Figure 3. Cytological smears showed numerous spindle-shaped sarcomatous cells exhibiting mild cyto-nuclear pleomorphism. Numerous naked nuclei and a hemorrhagic background were also found. Histological sections showed a typical angiosarcoma, rich in vascular spaces bordered by spindle-shaped sarcomatous cells.

RIBA should be differentiated from atypical vascular lesion (AVL), a cutaneous vascular lesion arising in patients who have received RT for breast carcinoma. This entity, first described by Fineberg and Rosen in 1994 [34], has been attributed to lymphatic obstruction secondary to surgery and/or RT [35,36]. A constant finding of AVL is a variable degree of chronic inflammatory infiltrate, and most of them resemble benign lymphangioendothelioma and/or lymphangioma circumscriptum. Although most authors consider AVLs as benign, a number of cases progressing to angiosarcoma have been reported, raising the concern of whether AVLs are premalignant or not [35,36,37,38]. Differential diagnosis between AVL and well-differentiated angiosarcoma can be extremely difficult in small biopsies [39]. Features that favor diagnosis of AVL include relative circumscription, bloodless spaces, and delicate projections of endothelial-lined stroma into vessel lumens, while infiltrative pattern, prominent dissection of dermal collagen, hemorrhage, extravasated red blood cells, blood lakes, and cytologic atypia are the characteristics of angiosarcomas [36].

## 3. Genetic Alterations in Radiation-Induced Breast Sarcomas

Some genetic factors have been associated with a higher risk of developing a radiation-induced sarcoma (RIS). In patients with Li–Fraumeni syndrome (TP53 mutation), an increased risk of BS following RT for breast carcinoma has been reported. A 33% risk of RIBS in patients with Li–Fraumeni syndrome and a history of breast carcinoma was reported by Heyman et al. [40], whereas a recent study found a much lower risk of 6% [41]. Alterations of the tumor suppressor gene TP53 were also identified in a large proportion of AVLs and RIBA, suggesting that the two entities are the extremes of a morphological continuum [42]. The TP53 gene is considered a guardian of the genome, as it is involved in DNA repair, the regulation of cell cycle checkpoints, etc. TP53 mutations have been associated with impaired repair of DNA damages induced by RT in some studies, and its use is avoided in patients with Li–Fraumeni syndrome [43].

A very limited number of RIBS cases in BRCA mutation carriers have been reported, raising the question of a possible association between BRCA mutation and secondary sarcomas [44]. BRCA proteins, encoded by the tumor suppressor genes BRCA1 and BRCA2, regulate DNA double-strand breaks repair. Ionizing radiation induce mainly double strand breaks of DNA, and BRCA mutations may lead to impaired radiation response, raising concerns about the use of RT in these patients [45]. However, the largest study of breast cancer patients with BRCA mutations treated with RT, which included 230 women in Israel (of whom 80% with an Ashkenazi Jewish founder mutation), found no increased risk of RIBS [46]. Thus, RT can be administered in BRCA-associated breast carcinoma.

Carriers of the retinoblastoma (Rb) germline mutation have a strong predisposition to cell cycle dysregulation, and an increased risk of STS post-radiation for retinoblastoma has been established, in a dose-dependent manner [47]. However, there is no data on the susceptibility to breast sarcoma. Similar to BRCA and TP53 mutations, concerns have been expressed about a deleterious role of RT in patients carrying Rb mutations [48].

In angiosarcomas in particular, recent comparative genomic hybridization studies revealed genetic differences between primary and secondary tumors. MYC gene amplification as a result of recurrent genetic alterations on chromosome 8q24.21 is highly indicative of secondary angiosarcoma [49,50,51]. However, there is a small subset of primary angiosarcomas harboring also MYC amplification [52]. Further research on MYC in secondary angiosarcomas, AVL and other RIS revealed high levels of MYC amplification by fluorescence in situ hybridization (FISH) only in secondary angiosarcomas, as opposed to AVL cases or AVL lesions adjacent to RIBA [51,53]. Moreover, incidence of MYC alteration was low in radiation-induced non-breast angiosarcomas [52], or absent in RIS without angiosarcoma morphology [54]. The incidence of MYC amplification, detected by FISH, in radiation- or lymphedema-induced angiosarcomas, ranges between 54% and 100% of studied cases [49,50,54]. A high concordance between MYC gene amplification and protein expression by immunohistochemistry (IHC) was reported in subsequent studies in secondary mammary angiosarcomas, AVLs, and primary mammary AS [50,55].

## 4. Pathophysiology

For RIBS, RT is likelythe main causative factor, although a clear dose–response relationship has not yet been established. Childhood high-dose fractionated radiation exposure (10+ Gy) has been linked to an increased risk of bone and STS, and a linear relationship was found between dose and risk [56]. Such a dose–response relationship has also been described in women receiving RT for BC [57]. According to an earlier report, a direct dose–risk relationship between RT dose and sarcoma risk was observed until 150–200 Joules [16]. Apart from the dose, the techniques of radiotherapy also seem to play a role in the development of secondary tumors in cancer survivors. Technological efforts during the last two decades aiming at a better radiation distribution and a reduction of normal tissue damage are expected to reduce the incidence of these neoplasms. The combination of RT and alkylating chemotherapy agents is also considered a risk factor for BS [58,59].

A large single-institution study of 13,472 patients who received RT for breast cancer at the Curie Institute reported a cumulative incidence of 0.48% for RIBS at 15 years [22]. In another large Canadian population-based study, the standardized incidence ratio was 26.2 for RIBA and 2.5 for RIBS [12]. The estimated incidence of RIBA after breast-conserving therapy ranges from 0.05% to 0.1% [15,60,61].

The clear role of RT in SBS reflects different hypotheses regarding its exact pathophysiological mechanism. The first hypothesis supports a direct effect of RT due to tissue damage [62]. On the other hand, a central role of lymphedema, as a result of lymphatic channels obstruction from RT or surgery, has also been speculated to be the principal causative factor of SBS [26]. In support of the second hypothesis, there are several reports of SBS in patients who underwent surgery for breast carcinoma without adjuvant RT [12,15], likely related to surgical sequelae, prior chemotherapy, and other environmental factors. In addition, a large Swedish study of secondary sarcomas post-treatment for BC found that the risk of STS, other than angiosarcomas, correlated with RT dose, whereas angiosarcomas risk correlated only with lymphoedema [16]. Furthermore, lymphangiosarcomas develop in patients with chronic lymphedema, even in the absence of RT [63]. Cases of out-of-field sarcomas have also been described in patients who underwent RT for BC [15]. Finally, radiation-induced angiosarcomas also occur in female reproductive organs after RT for previous cancers, raising the question of a possible association with female hormones [64].

## 5. Clinical Presentation of Radiation-Induced Breast Sarcomas

RIBS represent a distinct entity compared to other radiation-induced STS. The latency period between RT and the diagnosis of SBS is typically shorter in RIBS than in other STS [65]. The median latency period varies between 4.9 and 8.8 years in the different retrospective series, whereas SBS diagnosis is not established earlier than one year post RT [10,11,22,25]. The risk of BS after RT reaches its peak at 10years and then, although it declines, remains elevated for more than 20 years [66]. Therefore, while patients with PBS are usually in their fifth or sixth decade of life at the time of presentation [2,6,30], patients diagnosed of RIBS are usually older (Table 1). RIBA is likely distinct from radiation-induced angiosarcoma of other locations, as depicted by the difference in the latency period in several studies, which is shorter in the former (approximately 7 years versus 10–30 years, respectively) [31,67,68]. A more recent comparative study reported a mean latency of 6.7 versus 20.9 years for breast and non-breast angiosarcoma, respectively [28].

The clinical presentation of RIBS is analogous to PBS. Most patients present with a unilateral, painless breast lump arising within the irradiated region, with a median diameter of approximately 5 cm in most series (Table 2). Irradiated fields may be more fibrotic or sclerotic, rendering physical examination and evaluation of the lesion more difficult. RIBA have distinct clinical findings, including discoloration, purplish-red nodules, thickening or elevation of the skin, and a diffuse pattern of extension. There is often a delay in diagnosis, as typical lesions mimic ecchymosis, eczema, and benign post-radiation skin changes [69]. Consequently, the window of opportunity for timely treatment can be lost. Multifocality is another common feature of RIBA, combined with microsatellite lesions, in some cases. An initial indolent phase corresponding to low-grade RIBA is possible, followed by a sudden local progression coupled with high-grade histology [33].

AVL, on the other hand, arises in irradiated skin as solitary or multiple well-circumscribed, small, flesh-colored/reddish papules, or small erythematous patches/plaques, with a mean diameter of 8 mm (1–60 mm). Usually, these lesions are limited to superficial and middermis, although cases extending into the deep dermis and subcutis have also been described [36,38]. In one study, angiosarcomas presented as larger lesions compared to AVLs (median diameter of 7.5 cm versus 0.5 cm) [70]. The latency period of AVL is shorter, ranging from 3.5 to 6 years [36,37,38,70].

Similar to BS in general, mammography and ultra-sound findings of RIBS are not specific. False negative results can be obtained in the case of breast angiosarcoma, since both post-RT, and angiosarcoma-related skin changes may be mammographically indistinct [71,72]. According to available data, around 33% of angiosarcomas are not detected in mammography [73]. MRI can accurately detect tumor extension. Pronounced skin enhancement is a constant MRI finding of RIBA [74]. A large retrospective study of RIBA [71] reported diffuse T2 high-signal skin thickening, with persistent enhancement and rapidly enhanced T1-weighted images in all patients. Seven out of sixteen patients showed nodular foci of rapid, early arterial enhancement with washout hypointense lesions on T2-weighted images. Only four patients displayed distinct intraparenchymal masses.

## 6. Management of Early Disease

As for all STS, the optimal management of RIBS should be discussed by a multidisciplinary team following biopsy, in order to plan the best therapeutic strategy. The cornerstone of the treatment of localized disease is surgery, while the beneficial effect of adjuvant therapies in resectable disease remains unclear. The tumor growth pattern and involvement of the surrounding tissues will determine the correct therapeutic strategy, taking into consideration the previous treatment modalities of these patients.

### 6.1. Surgery

Positive surgical margins have been consistently reported to have a detrimental impact in overall survival [4,23,30,75] and a higher risk of local recurrence [76], turning this into the most important prognostic factor. Closer margins have also been associated with a higher risk of local recurrence when compared to wider margins [76], although this issue may be discussed, since McGowan et al. [30] concluded, after the analysis of a large breast sarcoma cohort with 26 RIS, that negative margins are adequate, irrespective of how close they are, but the tendency for satellite deposits of RIBA suggests that margins of 2 to 5 cm are preferable [33,77,78].

Prior RT-related tissue changes and the diffusively infiltrative margins of angiosarcoma make the complete excision of the tumor a surgical challenge, mandating the performance of surgery at experienced, high-volume centers. Pencavel et al. [3] retrospectively analyzed a large series of breast sarcomas, including both PBS and SBS, and indicated a better prognosis when the primary surgery was performed in a high-volume tertiary sarcoma center, when compared to patients who were referred to complete their treatment after a primary surgery with margin involvement. Unfortunately, these data support that wide local excision (WLE) may only be an option for low-grade tumors of up to 5 cm of diameter. Τhe excision of all irradiated tissues should be preferred, and mastectomy more often achieves negative margins compared to WLE [78]. In most series, the majority of patients underwent mastectomy [3,22,31,32,78,79,80], providing a robust base of data for recommending this approach.

As with other STS, BS rarely metastasize to regional lymph nodes. In most series, only anecdotal cases of nodal metastases are reported [81]. According to a large Surveillance, Epidemiology, and End Results (SEER) database study, approximately 40% of breast sarcoma patients undergo some degree of lymphadenectomy, with only 6/246 positive cases [82]. In most studies with RIBS, the incidence of nodal metastasis is not mentioned or not investigated, while those reporting any regional lymph-node involvement corroborate the low incidence indicated by the SEER. However, only one series of 20 RIBA patients from Brazil [83] reports a higher frequency of axillary nodal involvement, with two patients initially diagnosed with axillary metastases and another four with axillary recurrence. The available data support that sentinel lymph node biopsy, or systematic lymph-node dissection, is not indicated as part of the treatment of RIBS.

Approximately one out of two patients will present local recurrence, despite an initial wide surgical treatment. Resection of local recurrence improves local control and survival, with an estimated benefit in overall survival of over two years (34 versus 6 months) [78].

### 6.2. Radiation Therapy

The natural history of RIBS, in conjunction with the fact that surrounding or underlying tissues have probably received the maximum tolerated dose of radiation, render clinicians reluctant to include radiation in the therapeutic plan. In contrast, the aggressiveness of RIBS and especially angiosarcomas, as manifested by the high local recurrence rate, fosters the interest in evaluating additional treatment modalities, such as RT. In most series, only a small percentage of patients with resected disease underwent adjuvant RT without being associated with a better prognosis (Table 3). The rationale of patient selection for RT is not always clear and depends on the local practice and the experience of each group. Therefore, we cannot draw any definite conclusions on the role of RT in the management of RIBS.

A limited number of studies have shown a meaningful clinical benefit of the use of adjuvant RT. A large systematic review analyzed the role of adjuvant RT in RIBA; 17% of the patients received reirradiation, which was associated with local recurrence-free survival (LRFS) prolongation (5-year LRFS 57% versus 34% in patients who did not undergo adjuvant RT) [84]. Modesto et al. reported a trend for OS benefit in patients who received adjuvant RT for RT-induced sarcomas in general [85]. Lastly, an improved prognosis was demonstrated by two studies that included both primary and secondary tumors with adjuvant RT in breast angiosarcomas in general [30,86]. On the contrary, a detrimental effect of reirradiation on survival emerged by multivariate analysis in a population-based USA cohort study using the SEER database [87]. Whether this finding reflects patients’ vulnerability to RT damage or is due to confounding factors has yet to be defined.

Hyperfractionated accelerated RT (HART) may be of particular benefit to RIBS. It has been evaluated as neoadjuvant or adjuvant therapy for secondary angiosarcomas. The small fractional doses and the moderate total dose, combined with the large field margins over a short period, provide a scientific rationale for its use in this type of highly proliferative tumor, with a good tolerance. Fourteen patients with RIBA received HART at the University of Florida, six after surgery and eight as the initial treatment [33,88]. The technique consisted of three RT treatments per day, with a dose of 1 Gy per fraction, and varying total doses depending on the risk for subclinical disease. In all seven patients that underwent surgery after HART, a pathologic complete response was noted. The outcomes of the whole group are very promising, with a median OS of at least 7 years, a 5-year OS of 86%, and an acceptable toxicity.

RT in combination with hyperthermia has also been proposed in the localized setting, in the case of both resectable and unresectable disease. Two small studies from the Netherlands evaluated this treatment modality in RIS and angiosarcomas of the chest wall, reporting an improvement in local control [80,89]. In the first study, the combination therapy was administered mainly in patients with unresectable disease with a response rate of 75%, whereas in the second case, the therapy was also delivered in the adjuvant setting. However, the median overall survival in these studies was 1–1.5 years.

### 6.3. Systemic Therapy (Adjuvant/Neoadjuvant Chemotherapy)

Similar to other STS, the role of adjuvant chemotherapy in breast sarcoma and RIBS, in particular, remains uncertain. In most retrospective studies, only a small number of patients, if any, received chemotherapy after complete surgical resection (Table 3), with no documented impact on survival [25,26,79,83,90]. However, there are some data supporting a benefit from adjuvant chemotherapy in angiosarcoma patients. Evidence of OS benefit was reported by Stanford, in a small study including 34 secondary breast angiosarcoma patients [91]. The survival benefit was not found in patients with primary angiosarcoma. A larger study of RT-induced angiosarcoma of the breast and chest wall from M.D. Anderson, in which approximately half of the patients were given perioperative chemotherapy (mostly adjuvant), demonstrated a lower risk for local recurrence with chemotherapy [31]. A small study of primary and secondary breast angiosarcoma reported a trend towards an improved RFS with chemotherapy, which was not confirmed in a meta-analysis of almost 1000 breast angiosarcomas [92]. Lastly, a recent nationwide analysis of breast angiosarcomas in the USA revealed a prolonged OS in patients with large tumors (>5 cm) who underwent adjuvant chemotherapy [93].

There is an obvious paucity of data regarding the role of adjuvant chemotherapy in RIBS. The current studies are small, and in the larger ones, only a small subset of patients received chemotherapy. Only a few studies separately analyze secondary breast sarcomas. Furthermore, the selection criteria forchemotherapy are not consistent in these studies, and only in a few cases are the criteria clearly defined. In the study from M.D. Anderson, for example, chemotherapy was used for large or high-grade tumors in the case of close surgical margins [31]. Likewise, in a single-institution study from Brazil, patients with tumors larger than 2 cm or of high-grade were treated with chemotherapy [83].

The data for neoadjuvant chemotherapy are even scarcer. It is definitely an option for locally advanced inoperable disease, with some theoretical advantages, including the potential downsizing of the tumor and the facilitation of R0 resection [72]. In a retrospective series of RIBA from Royal Marsden, neoadjuvant chemotherapy was delivered in the case of seven unresectable tumors, of which three were finally operated and two had pathological complete response (pCR) [76]. In addition, the evidence on the role of neoadjuvant chemotherapy comes from a limited number of case-reports [94,95,96]. It should be noted that histotype-tailored chemotherapy regimens should be used, which was the case in the above reports. Angiosarcoma, in comparison to other STS histotypes, seems to be more chemosensitive, rendering neoadjuvant chemotherapy an acceptable approach.

## 7. Management of Inoperable/Metastatic Disease

### 7.1. Chemotherapy

Chemotherapy represents the standard first-line treatment in the advanced setting. There is very limited evidence on the activity of the different drugs in RIBS in particular, with the exception of some reports on breast angiosarcoma, and most data are extrapolated from STS trials. Anthracyclines monotherapy, or in combination with ifosfamide, constitute the mainstay of treatment in metastatic STS. Sher et al. reported an overall response rate (ORR) of 48% in 29 metastatic breast angiosarcomas to first-line anthracycline-ifosfamide or gemcitabine-taxane cytotoxic chemotherapy combination, concluding that the disease is quite chemosensitive [79]. It should be noted, however, that anthracyclines would likely have been used already for early breast cancer, limiting its administration to the maximum cumulative dose.

Paclitaxel is considered active against angiosarcomas and breast angiosarcoma in particular, making it an important treatment option after anthracycline, or even in the first-line setting. In the ANGIOTAX study, a small French phase II study of 30 angiosarcoma patients, of which 33% had breast angiosarcoma, the PFS at four months was 45% [24].Weekly paclitaxel was compared with doxorubicin as a fist-line treatment in a retrospective study of 117 metastatic angiosarcomas, of which 36 had secondary and 28 had cutaneous angiosarcoma [97]. The two drugs were found equal regarding PFS. Cutaneous angiosarcoma exhibited a higher response rate to paclitaxel compared to non-cutaneous angiosarcoma, and radiation-induced angiosarcomas exhibited more objective responses to both drugs when compared to primary tumors. According to a retrospective series that evaluated RIBA specifically, the median treatment time (MTT) on first-line paclitaxel was 3.5 months [98].

Both of the above drugs show dose-limiting toxic effects (cardiac and neurological). Gemcitabine is another chemotherapeutic agent that can be used in the treatment of RIBS. It was evaluated retrospectively as single agent in 25 angiosarcomas from the Italian Rare Cancer Network (including eight radiation-induced angiosarcomas and seven RIBA). The ORR was 68%, the median PFS was 7 months, and the median OS was 17 months [99]. Six out of eight radiation-induced angiosarcomas responded to the drug. Gemcitabine can also be combined with docetaxel, with an increased ORR, although there is no data on breast sarcoma in particular. In addition, liposomal pegylated doxorubicin, trabectedin, and navelbine are other therapies that can be used, based on extrapolated evidence from STS trials.

### 7.2. Bevacizumab

Vascular endothelial growth factor (VEGF) and its receptor (VEGFR) are expressed in breast angiosarcomas [100] and angiosarcomas in general [101], and they likely play a significant role in carcinogenesis and angiosarcoma progression. VEGFR expression was associated with grade 1 or 2 angiosarcoma of the breast [100], whereas specific data regarding RIBA (usually a grade 3 tumor) are lacking. Bevacizumab is a monoclonal antibody targeting VEGF that is approved for the treatment of different tumors. The activity of bevacizumab monotherapy was demonstrated in a small study of epithelioid hemangioendothelioma and angiosarcoma, including four patients with breast angiosarcoma [102]. Of the 30 patients, 15 exhibited stable disease and four had a partial response. The combination of bevacizumab and paclitaxel was also compared to paclitaxel monotherapy in a randomized phase 2 study of 49 angiosarcoma patients, of which 24 had a primary breast tumor and 24 had a history of RT [103]. The combination was more toxic and did not improve efficacy, although there was no separate analysis for breast sarcoma.

### 7.3. Tyrosine Kinase Inhibitors

In addition to VEGF/VEGFR, other receptor tyrosine kinases (RTK) have been shown to be expressed or upregulated at the mRNA level in some STS, such as PDGFR [104]. Pazopanib is a small multi-targeted tyrosine kinase inhibitor (TKI) against VEGFR1, VEGFR2, VEGFR3, and PDGFR, which is approved for the treatment of metastatic STS after anthracycline-based chemotherapy, or in the first line of treatment in patients not eligible for this therapy [105]. A retrospective EORTC study of pazopanib in advanced vascular sarcomas revealed an ORR of 20% in 40 angiosarcoma patients, which was similar in radiation-induced and non-radiation-induced angiosarcomas [106].

Sorafenib is another multikinase inhibitor targeting Raf, PDGF, VEGFR2, VEGFR3, and c-Kit. In a phase 2 trial of STS, sorafenib was found active only against angiosarcoma, with 5/37 showing at least partial response [107]. Three of the five patients who responded to treatment had RIBA, and the remaining two had angiosarcoma of the head and neck. However, in the phase 2 S050 trial, which evaluated sorafenib in selected STS histotypes, including five angiosarcomas, no responses were noted according to RECIST criteria, while a clinical benefit was observed in the majority of angiosarcoma patients [108]. The first study, from Memorial Sloan Kettering Cancer Center (MSKCC), that evaluated systemic therapies specifically in RIBA, used median treatment time (MTT) on the first line as an outcome measurement and found the longest MTT of 25.1 months with sorafenib [98]. This result may be due to MYC and FLT4 co-amplifications harbored by some RIBA [109].

A high positivity for Kit RTK by IHC has been demonstrated in a small study of RIS, including RIBAs, which is not coupled with exon 11 mutations [110]. Whether Kit expression drives oncogenesis or not remains unknown. However, this observation provides a rationale for imatinib testing in RIBS.

### 7.4. Immunotherapy

Immunotherapy (IO) constitutes a novel approach in the armamentarium against cancers. The immune profile of RIS was explored in a small study of 20 cases, including three angiosarcomas [111]. A higher mutational rate was found, compared to other cancer types, coupled with 45% positivity to PD1/PDL1 and tumor-infiltrating lymphocytes, whereas primary breast angiosarcomas exhibit a lower PDL1 expression [112]. In most studies of immune checkpoint inhibitors (ICI), as a monotherapy or in combinations, for the treatment of STS, breast sarcomas were underrepresented. It is worth mentioning that there are some isolated responses reported in angiosarcomas, without information on tumor location or radiation history [113,114]. In a small retrospective series of angiosarcomas, responses to ICI have been reported for seven patients, including a patient with RIBA [115].

## 8. Prognosis

Prognosis of breast sarcomas is highly dependent upon histotype, histological grade, and tumor size [25,30,31,76,116]. The extent of surgery and margin status is also an important prognostic factor. In general, RIS are considered as aggressive tumors. In a large multivariate analysis that adjusted for age, tumor size, depth, and margin status, it was demonstrated that RIS are associated with inferior survival rate compared to sporadic STS, and previous radiotherapy is an independent prognostic factor [117]. For these patients, previous treatments and subsequent side effects (i.e., bone marrow dysfunction) also reduce therapeutic options.

Furthermore, the high frequency of angiosarcoma histology among RIBS partially accounts for their aggressive behavior. Both primary and therapy-related angiosarcomas have been associated with poor prognosis. Although some studies report a worse prognosis of RIBA compared to PBA [83,86,87], it has not been confirmed in all series [32,66,81]. In most of these studies, survival endpoints have been compared between primary and secondary breast angiosarcomas without analyzing potential confounding factors, such as tumor grade and stage. Only the USA comparative cohort study from the SEER database adjusted for prognostic characteristics of patients and concluded that the worse prognosis of RIBA was due to advanced stage, grade, and age [87].

More specifically, in the older studies, the median OS for RIBA was approximately two to three years [78,80,83] (Table 3). Early local recurrences are reflected by a short LRFS of 1.29 year in the MSKCC series, in which tumor depth was associated with RFS, and one out of four patients with local recurrence died of the disease [98]. However, there are recent data from large series reporting longer survival. A median OS of five years was reported in a large UK study [81]. According to a recent study analyzing a national database from the Netherlands, 5-year and 10-year OS in RIBA patients was 40.5% and 25%, respectively [61]. A higher 5-year OS of 63.5% was found in a study from Stanford, in which most patients received total mastectomy plus wide excision of the skin [91]. This improved survival rate is likely due to the increased awareness of RIBA and the multimodal management of sarcomas.

## 9. Conclusions and Perspectives

Radiation-induced breast sarcomas represent an iatrogenic disease with distinct characteristics, such as a shorter latency period compared to other RIS. Although they are very rare, their incidence has increased in recent years, most likely due to the establishment of breast-conserving surgery in conjunction with adjuvant RT for the treatment of BC. Notwithstanding, their incidence is still so low that the benefits of RT for BC outweigh the risk for SBS. The improvement of radiation techniques in the future is warranted to prevent this disease. RIBS are, in general, aggressive tumors with poor prognosis, driven at least partially by the high incidence of angiosarcoma histology and grade 3 tumors. RIBA is the most common subtype of RIBS and has unique histological, molecular, and clinical features.

For localized disease, early diagnosis allowing for proper surgery with negative margins is the only chance for cure. Given the high rate of local recurrence and the challenging surgical management in RIBA, including the assessment of tumor margins, we believe that an aggressive approach, i.e., mastectomy, is indicated. In addition, the poor outcomes, even in series with negative margins, may suggest that surgery alone is not curative, and maximum treatment should be offered to RIBA patients. There is limited evidence for LRFS prolongation with adjuvant RT. Improved global outcomes have been reported with HART, making it an interesting treatment option to evaluate further. Despite the very limited data on adjuvant/neoadjuvant chemotherapy, we recommend its use in fit patients. Randomized trials through international collaboration assessing the role of adjuvant therapies should be a priority.

In the advanced setting, there are limited therapeutic options, most of which have not been tested in RIBS or RIBA patients. The need for histotype-tailored therapies, as well as personalized therapies based on molecular profiling, has emerged in the management of STS. Next-generation sequencing data on RIBS and RIBA are required, as no molecular candidates have yet been recognized to guide targeted treatments. For example, a small study demonstrated that 10% of patients with primary and secondary angiosarcomas uniquely to the breast exhibit activating mutations in KDR (also known as VEGFR2) [118]. VEGFR inhibitors could be studied in these tumors, based on their activity in vitro. In addition, immunohistochemical studies in angiosarcomas, including both RIBA and primary breast angiosarcomas, have shown activation of the PIK3CA/Akt/mTOR pathway in an important subset of the disease [97,116]. Therefore, mTOR and PIK3CA inhibitors may be relevant targeted therapies to explore. The mTOR inhibitor sirolimus, in combination with cyclophosphamide, was evaluated in a small study of different STS, leading to SD in approximately 20% of patients [119]. In the study from MSKCC, one RIBA patient received sirolimus, with a MTT of 14.9 months [98]. Finally, the relatively high levels of immunotherapy biomarkers in RIS should be further investigated in larger studies, specifically in RIBS. If these results are confirmed, immune checkpoint inhibitors or other IO approaches should be tested in clinical trials.

To conclude, the current therapies which are beneficial for RIBS are very limited, both in the early and the advanced settings. There is an urgent medical need for new therapies to reversethe current dismal prognosis.

## Figures and Tables

**Figure 1 ijms-23-04125-f001:**
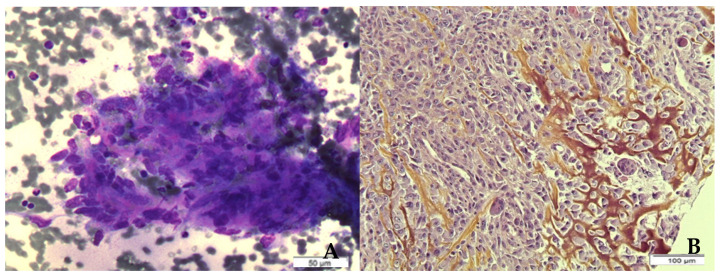
Radiation-induced osteosarcoma. (**A**): smears showing clustered spindle and pleomorphic sarcomatous cells associated with pinkish/orange osteoid (May-Grunwald–Giemsa, 400×; Merck, Darmstadt, Germany); (**B**): corresponding surgical sections of spindle-shaped and epithelioid proliferation showing osteogenesis (Hematoxylin-Eosin-Safran,200×; Merck, Darmstadt, Germany). Schemes follow the same formatting.

**Figure 2 ijms-23-04125-f002:**
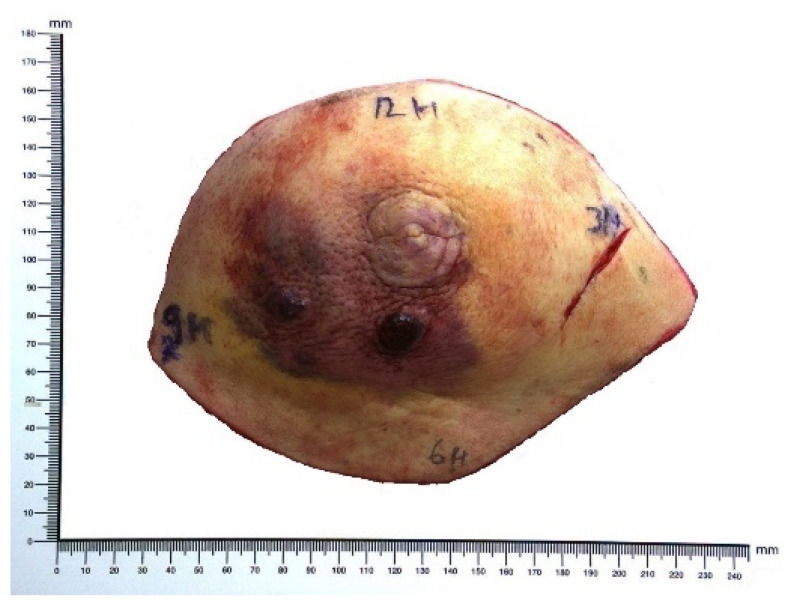
Surgical mammary specimen of radiation-induced breast angiosarcoma. Note violet, congestive skin and subcutaneous tumoral infiltration. Morphological cytologic and histologic findings are shown in Figure 3.

**Figure 3 ijms-23-04125-f003:**
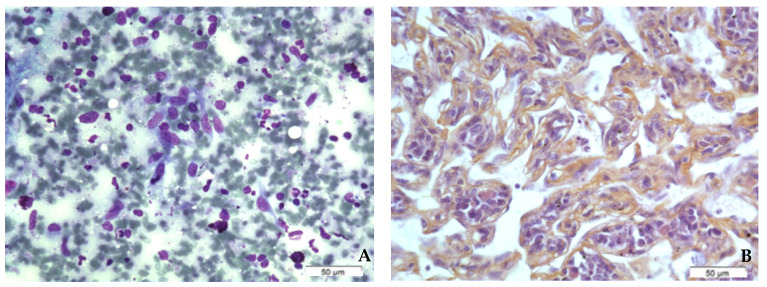
The same case of radiation-induced breast angiosarcoma from the Figure 2. (**A**): cytologic smears showing numerous spindle-shaped sarcomatous cells exhibiting mild cyto-nuclear pleomorphism. Note the presence of naked nuclei and hemorrhagic background (May-Grunwald–Giemsa 400×; Merck, Darmstadt, Germany). (**B**): Corresponding surgical sections showing typical angiosarcoma rich in vascular spaces bordered by spindle-shaped sarcomatous cells (Hematoxylin-Eosin-Safran, 400×; Merck, Darmstadt, Germany).

**Table 1 ijms-23-04125-t001:** Clinicopathological differences between primary and radiation-induced breast sarcoma.

	Primary Breast Sarcoma	Radiation-Induced Breast Sarcoma
Frequency	Rare	Rare
Age	5th–6th decade	Depends on first cancer age and latency period
Risk factors	Unknown, genetic predisposition	Young age of RT, long latency period, high radiation dosage, alkylating agents, genetic predisposition
Clinical presentation	Unilateral breast lump	Unilateral breast lump, discoloration, purplish-red nodules, thickening or elevation of the skin, and a diffuse pattern of extension
Histology	UPS, FS, AS	AS
Prognosis	Poor	Poor

**Table 2 ijms-23-04125-t002:** Clinicopathological findings of retrospective studies including ≥10 cases of radiation-induced breast sarcoma (excluding bone sarcomas). The mean value of age at diagnosis and primary tumor diameter (T) is reported instead of the median, when the latter is not available. NA: Not available.

Author	Year of Publication	Treatment Period	Total N	AS (%)	Median Age (Years)	Latency Period (Years)	Median T (cm)
Karlsson	1998	1958–1992	67	47.8	NA	NA	NA
Lagrange	2000	1975–1995	14	42.9	65.5	NA	NA
Blanchard	2002	1975–2001	34	35.3	62.3 (mean)	12.7 (mean)	NA
Billings	2004	<2004	27	100	70	4.9	4
Kirova	2005	1984–2005	18	72.2	66.5	7.3	NA
Sher	2007	1965–2002	13	100	72	7	NA
Hodgson	2007	1981–2000	31	100	72.9 (mean)	5.2 (mean)	NA
Palta	2010	1997–2006	14	100	66.5	7.7	ΝA
Pencavel	2011	1996–2006	19	78.9	61 (mean)	ΝA	NA
Seinen	2012	1990–2009	35	100	67	7	4
Fraga-Guedes	2012	1999–2009	20	100	66	7.5	2.8
Torres	2013	1993–2011	95	100	71	7	5
Linthorst	2013	2000–2011	23	100	70	8.8	NA
D’Angelo	2013	1982–2011	79	100	68	7	4.2
Cohen-Hallaleh	2017	2000–2014	49	100	72	7.5	5
Gervais	2017	1994–2014	20	100	71	8	5–10
Yin	2017	1973–2012	173	100	70–74	NA	NA
Abdou	2019	1990–2015	13	100	71	7.8	6.9
Rombouts	2019	1989–2017	209	100	73	8	NM
Gutkin	2020	1998–2019	34	100	72	6.9	5.6 (mean)

**Table 3 ijms-23-04125-t003:** Treatment modalities and prognosis of radiation-induced breast sarcoma in retrospective studies including ≥10 cases (excluding bone sarcomas). NA: Not available.

Author	Year of Publication	Nodal Involvem.	Type of Surgery (Ν)	Margin Status (Ν)	Adjuvant RT (%)	(neo)Adjuvant Chemo (%)	OS/DFS (Years)	Prognostic Association
Karlsson	1998	NA	NA	NA	NA	NA	NA	no
Lagrange	2000	NA	2 MA, 8 WLE	2 R2	28.6	35.7	NA	surgery
Blanchard	2002	NA	30/34 surgery	NA	30	43	NA	size
Billings	2004	NA	10 MA, 10 WLE	NA	10	20	ΝA	no
Kirova	2005	NA	11 MA, 5 WLE	NA	5.6	5.6	mOS = 22 m	no
Sher	2007	ΝΕ	12 MA, 1 WLE	ΝA	0	NA	NA	size
Hodgson	2007	NA	25 MA, 1 WLE	NA	0	NA	NA	no
Palta	2010	2/14	14 ΜA	NA	100 (HART)	0	5y-OS = 86%, 5y-PFS = 64%	benefit of HART in addition to surgery
Pencavel	2011	0/3	12 MA, 6 WLE	ΝA	ΝA	ΝA	mDFS = 30 m. 5y-DFS = 26%	surgery at experienced center
Seinen	2012	NA	24 MA, 7 WLE	23 R0, 1R1, 7 R2	3.2	3.2	mDFS = 16 m	amenable to surgery for local recurrence
Fraga-Guedes	2012	0	15 MA	ΝA	10	50	5y-OS = 28.2%	grade, prior RT
Torres	2013	0	60 MA, 27 WLE	81 RO, 4 R1, 4 R2	0	52	5y-OS = 91%	size
Linthorst	2013	NA	10 MA, 1 WLE	4/11 R0, 6/11 R1, 1/11 R2	34.8	0	mOS = 18 m	reRT + hyperthermia (local control)
D’Angelo	2013	NA	65 MA, 13 WLE	45 R0, 12 R1, 8 R2	ΝA	11.4	mDSS = 3 y.	age > 68 y, depth
Cohen-Hallaleh	2017	NE	38 MA	32/37 R0	0	19.1	mOS = 37 m (resectable)	size, resectability
Gervais	2017	NA	19 MA, 1 WLE	18 R0	35	50	mOS = 51 m	no
Yin	2017	NA	NA	NA	12.7	NA	mOS = 32 m	age, tumor spread
Abdou	2019	NA	9 MA	NA	7.7	61.5	mOS = 64.2 m	no
Rombouts	2019	NA	ΝA	ΝA	9.1	1.4	5y-OS = 40.5%	no
Gutkin	2020	0	27 MA, 4 WLE	12 R0, 6 R1	8.8	44.1	mOS = 16.9 y	chemotherapy

## Data Availability

Not applicable.

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
