# Peer review of "Clinical and Molecular Insights of Radiation-Induced Breast Sarcomas: Is There Hope on the Horizon for Effective Treatment of This Aggressive Disease?"

_ijms, 2022, doi:10.3390/ijms23084125_

Round 1

Reviewer 1 Report

The review article “Clinical and molecular insights of radiation-induced breast sarcomas: is there hope on the horizon for effective treatment of this aggressive disease?” is well-structured and covers a low incident but relevant topic. The clinical and histopathological characteristics of RIBS are discussed, as well as their management and prognosis, emphasizing their distinct characteristics versus primary breast sarcomas.

Some minor general points to be addressed:

The references cover the last 20 years (or more). Authors should review if references older than 2000 are needed.

Figure 2 is allowed to be published? Is any ethical letter needed? The X-axis lacks legend. The red blood should be removed. The numbers within the circles are not clear.  The image quality should be 300 dpi.

The atypical vascular lesion (AVL), should be defined the first time appearing.

The abstract does not cover the content of the manuscript.

Author Response

Thank you very much for your time reviewing this manuscript and your comments. We replied to all of them and modified the manuscript accordinlgy.

Reviewer 2 Report

Kokkali et al. comprehensively review the scientific literature surrounding radiation-induced breast sarcomas and discuss the effectiveness of potential treatments for RIBS. Overall, the review is well organized and thorough. I have minor comments that should be addressed before this review is accepted for publication. 

  1. Bars must be included in both images in Figure 2.
  2. In section 3 where the authors discuss RIBS in BRCA-mutation carriers, a brief statement should be added on the function of BRCA proteins in genomic instability and potential concerns regarding the use of radiotherapy in these patients. Similar changes should be included when discussing P53 mutations in Li-Fraumeni syndrome and other genetic alterations. 
  3. PTEN mutations occur at a high frequency in breast cancer patients. Is there a link between PTEN and RIBS? 

Author Response

Thank you very much for your time reviewing this manuscript and your comments. All your points are answered in the attached document and modifications were made accordingly in the manuscript.
